# Disaster Preparedness among Service Dog Puppy- Raisers (Human Subject Sample)

**DOI:** 10.3390/ani10020246

**Published:** 2020-02-04

**Authors:** Sarah E. DeYoung, Ashley K. Farmer, Zoe Callaro, Shelby Naar

**Affiliations:** 1Disaster Research Center, Department of Sociology and Criminal Justice, University of Delaware, Newark, DE 19716, USA; 2Department of Criminal Justice Sciences, Illinois State University, Normal, IL 61761, USA; akfarme@ilstu.edu; 3Public Administration, University of Colorado Denver, Denver, CO 80204, USA; zoe.callaro@ucdenver.edu; 4Radiological Emergency Preparedness, Georgia Emergency Management and Homeland Security, Atlanta, GA 30316, USA; shelby.naar@gema.ga.gov

**Keywords:** disaster, preparedness, puppy raiser, service dogs, working dogs

## Abstract

**Simple Summary:**

Before service animals are matched with clients, they undergo training programs for increasing the dogs’ ability to navigate public spaces. Increasingly, service dog training programs recruit young adults from universities and college campuses. Little is known, however, how these students prepare for disasters and the ways in which they create plans to keep service dogs in training safe during hazard events. We collected data from service dog puppy raisers in a hurricane-prone region of the United States to understand their concepts and actions of disaster preparedness. People who were raising a service puppy for the first time were more likely to consider evacuating from Hurricane Irma in 2017 than people who had participated in the program before. Additionally, over half of the respondents did not have a disaster preparedness kit. Finally, many respondents in this study indicated that their service dog in training provides a sense of safety and security.

**Abstract:**

Little is known about the ways in which puppy raisers engage in disaster preparedness for their puppies (or “guide dogs in training”). The aim of this research is to understand disaster preparedness among service dog puppy raisers. A web-based survey was distributed to people raising puppies in a service dog training program (*n* = 53 complete survey responses). Questions in the survey included items about disaster preparedness and plans for canine safety in hazards events. Out of those who said they had an evacuation plan for their puppy in training, 59% stated they would put the dog in their vehicles for evacuating to safety in the event of a hurricane or other disaster. The odds of first-time puppy raisers who considered evacuation for Hurricane Irma in 2017 was 15.3 times the odds of repeat raisers. Over half the raisers reported that they did not have a disaster kit. Additionally, 82% of respondents indicated that having a service puppy in training makes them feel safer. These results can be used as a foundation for service dog organizations in disaster preparedness among their puppy raiser volunteers and in designing recruitment messages for new volunteers.

## 1. Introduction

The Pets Evacuation and Transportation Standards (PETS) Act was passed on 6 October 2006 after category five Hurricane Katrina devastated the coast of Louisiana and Texas. This new act amended the Robert T. Stafford Disaster Relief and Emergency Assistance Act to address the needs of household companion animal and service animals [1]. However, preparedness information typically disseminated to communities through outreach programs or county emergency managers encourages companion animal guardians to prepare not only for themselves but also for their animals [2]. Individual preparedness—such as being able to care for household animals in the event of an emergency—helps reduce losses in disasters and puts less strain on first responders and resources [3]. Although the PETS Act was an important policy for creating plans for managing pets in disasters, individual preparedness remains extremely important for all companion animal guardians, service dog users, and those with service dogs in training [4].

Globally, pets and companion animals are a critical part of the human landscape. According to 2016 companion animal guardianship statistics, out of Australia’s 9.2 million households, 5.7 million housed a companion animal [5]. Companion animal guardianship has grown tremendously among the United States population, serving many different forms of companionship. According to the 2017–2018 American Companion Pet Products Association (APPA) National Companion Pet Guardians Survey, 68% of households within the U.S. own a pet. For context, 68% is equivalent to about 84.6 million homes [6]. The National Pet Survey was first conducted in the year 1988, where it found that about 56% of households owned a companion animal at that time. Besides the typical dog and cat, the survey now includes numbers for birds, horses, freshwater fish, saltwater fish, and even reptiles. In 1994, APPA calculated total U.S. companion animal industry expenditures to be $17 billion [6]. In 2016, that number soared to almost $67 billion, showing the bond between guardian and companion animal has strengthened over the past few decades [6].

While expenditures alone may not capture the overall social landscape of humans and their pets, news media coverage of humans and their pets in disasters is present in almost every disaster [7]. The ongoing concern about pets in disasters reflects the notion that pets are important in emergencies. Less attention has been given to the topic of service animals in disasters and the ways in which people who rear service animals engage in disaster preparedness. In the United States, college students are increasingly volunteering to train guide dog puppies—and many of the campuses in which they live are prone to hurricanes and other hazards. This research is an attempt at understanding the ways in which guide dog trainers prepare their puppies for disasters and emergency scenarios. Specifically, this research focuses on the emerging population of college student puppy raisers and the ways in which they perceive preparedness, safety, and risk for themselves and their puppies in training. We begin the paper by describing the context of the human-animal bond because this may influence decision-making such as disaster preparedness. We then describe research related to college students and the ways in which they interpret risk, followed by literature on the social landscape of animals in disasters. We then provide the method for our study followed by the results, discussion, and implications for future research.

### 1.1. Human-Animal Bond

Companion animals provide a multitude of benefits for their guardians. Although emotional benefits may be most obvious, companion animals can also have social, physiological, and physical health benefits [8,9,10]. Companion animal guardianship directly affects cardiovascular health, increases physical activity, enhances the welfare of the elderly, and even serves as a source of support for people experiencing crisis and hardship. These hardships can include homelessness, family stressors, or some type of disaster [9].

Therapeutic organizations have used animals in several areas, including animal-assisted therapy, animal-assisted crisis response, and child psychotherapy [8,11]. The role of pets is crucial, as reflected in some reports, where companion animals were the main reason individuals survived personal crises such as loss of a loved one, a family health crisis, or periods following substance abuse [9]. Benefits created by companion animals are not unique to one specific age, gender, or ethnicity; evidence suggests companion animals to help hospitalized children recover, ease suffering for those in hospice care, and alleviate depression among Acquired Immune Deficiency Syndrome (AIDS) patients [11].

Dogs, although popular companion animals, are also popular in occupational roles. The roles include guiding for the blind, search and rescue, assistance for people with disabilities, and police detection work [12,13]. Despite working dogs serving a different role than companion animals, they may have similar attachment levels to their handlers as pets do to their guardians [14]. Assistance dogs serve important roles for their handlers, especially for handlers who cannot see or are visually impaired [12].

### 1.2. History and Context of the Guide Dog Occupation

The origins of the guide dog can be traced back to Germany after the end of World War I and have now become more and more common throughout the U.S. and countries around the world [15]. Guide dogs play an important role for their handlers, serving as companions, security detail, and safety officers. After World War I, many German troops returned home with no physical injuries besides the loss of their eyesight [14]. In the U.K., many guide dog users expressed social and psychological benefits from having a guide dog, as well as increased confidence and socialization with other people [16].

The way other people treat those who are visually impaired has also been stated as a benefit among guide dog users; out in public people tend to be friendlier towards those individuals when the dog is present [16]. This particular benefit has been very important because individuals with visual impairment are already prone to social stigma and even discrimination [17]. In South Africa, guide dog users explained the dogs were safer and faster than using a cane, they provided more confidence regarding mobility, and enhanced independence [17]. Japanese guide dog teams expressed that their dogs reduced tension and strain when traveling and had positive effects on mobility and quality of life [18]. Many of these benefits originate from the strong bond and attachment that forms between handler and dog. Additionally, research on guide dog training programs suggests that existence of service animals can be linked with the perceived validity of a persons’ functional and access needs status [19].

### 1.3. Guide Dog-Handler Bond

Guide dogs go through many bonding stages during their training and placement process with a handler. They live and stay with a puppy raiser, which is typically a volunteer through the guide dog school, get assigned to a trainer once they are around sixteen months old, and then are eventually matched with a person who is visually impaired. Each of these stages has been isolated and studied to understand the difference in bond strength between the dogs and their raisers, the dogs and their trainers, and the dogs and their handlers [20,21]. For example, after one year of partnership, guide dogs showed all signs of attachment when separated from their handler, even in the presence of another friendly individual, and remained oriented to the door for their handler to return [20,21].

Some evidence suggests that because handlers who are blind have different situational awareness, some dogs developed signals to more effectively communicate with their handler, such as maintaining physical contact [21]. Because of the process used to train guide dogs, bonds are formed and technically lost after the dogs leave their puppy raisers and then again when they leave their trainers. However, the use of puppy raisers is still important in ensuring the dogs reach their end goal of helping individuals dependent upon their services [22].

### 1.4. Puppy Raisers

Guide dog schools utilize volunteer puppy raisers as an essential part of their process [22]. Puppy raisers typically have the dogs from around eight weeks old until a little over a year old or until the dog has matured. Because the puppy raisers have the dogs for quite a while, it is understandable that bonds form between the trainer and puppy. Puppy raisers can help contribute to a dog’s success or more accurately predict the ability of the dog they are raising to become a guide dog. For example, assessments or questionnaires done by puppy raisers concerning their dogs’ chances of success have served as important predictors for the respective guide dog schools [22,23,24]. This suggests that puppy raisers bond with and know their dog very well over just a year of time. Other benefits can also come with being a puppy raiser, including improved mental, social, and physical health; increased awareness of responsibility; and new friendships with other raisers [25].

Depending on the reach and location of each guide dog school, a diverse array of people volunteer as puppy raisers. In many instances, college students tend to be the most common group of volunteers. To become successful guide dogs, raisers take the puppies with them everywhere they go so that the dogs grow accustomed early on to what they will be doing as working canines. For working adults with full-time jobs, it is harder for them to be accompanied by young puppies every day, and the puppies are also big commitments. College students can more easily take a puppy in training to class, grocery shopping, or to social events [26]. Interestingly, the popularity of having a guide dog puppy is expressed as a trend among college students in some areas of the United States, like fashion and technology branding [27] that signifies identity, social status, and possible wealth. In fact, it is common to see students walking on campuses with the training dogs that wear the signature yellow vests. For the broader puppy-raiser population, training of the dogs is likely to take place by able-bodied persons [28]. In the context of this research, it is important to consider how the able-bodiedness of the handlers influences their ability to engage in activities such as disaster preparedness, as well as perceptions about ease of preparedness by able-bodied handlers [29].

### 1.5. College Students: Risk Perception

Risk perception is defined as a “highly personal process of decision making, based on an individual’s frame of reference developed over a lifetime” [30] (p. 1) and can be directly linked to protective actions taken [30,31] In other words, perception of risk is a very individualistic concept and will differ from person to person [32]. This important aspect of risk perception becomes a challenge in many risk communication strategies, particularly when risk perception is low.

Source and repetition of information also contribute to decreased risk perception. If the source is not trusted or people do not trust current science or technology in general, there will be lower general risk perception [33]. If information regarding a specific risk has become the social norm and is common knowledge, risk communicators will have a hard time heightening risk perception when necessary [34,35]. A major example of this would be safe driving practices for younger drivers (i.e., college students), which is a very popular risk communication topic. Adults between the age 18–29 are more likely to drive while drowsy compared to other age groups, and drivers in their 20’s make up 27% of distracted drivers in fatal crashes; however, college students have grown accustomed to car-related risk information and accept it as common knowledge [36,37]. Other factors contributing to low risk perception include optimistic bias, self-efficacy, perception of benefits, and perceived barriers to preventative action [35,38,39].

In general, higher perceived severity of a risk (understanding the risk has serious consequences) will lead to higher risk perception and cause an individual to engage in more physical protective behavior [40,41]. Higher perceived severity is common in those who have personally experienced certain risks in their area (i.e., tornadoes, earthquakes, wildfires), so it is understandable that heightened risk perception differs from region to region [42,43]. Among college students, local hazards, as well as a risk being uncontrollable or unknown, will contribute to higher risk perceptions [43].

### 1.6. College Students: Risk Information

The sources of risk information have changed as technology has advanced over the years [44]. For example, in 2010 students were more likely to receive information from family, friends, and television and less likely to receive it from university flyers, professors, or course content [45]. However, in 2012 students preferred to receive emergency preparedness information directly from their school via texts and emails [44]. These results were proven a few years prior when a tornado touched down near the Mississippi State University campus, and most of the students stated they first heard of the tornado through university alert messages [46]. This may be because students indirectly expect their university to “take care of them” in the event of an emergency or disaster [42].

### 1.7. Service Animals & Disasters

Because there were multiple instances of people being forcefully separated from their animals during Hurricane Katrina [47] there has been growing attention on the issue of animals and disasters. While the PETS Act was passed in 2006, the Act has limitations in terms of breed, disaster phases (e.g., recovery and long-term housing is not considered in the PETS Act, such as rental fees for pets of displaced persons), and refusal of evacuation and reunification issues remain a concern in disasters [48]. Simultaneously, the body of research on pets and disasters is growing– especially in the areas of understanding attachment [49,50] and issues related to grief [51,52]. There is also a growing body of research on the needs of functional-access individuals in disaster planning [53,54]. However, there is a scarcity of research on guide dog trainers and disaster planning and preparedness. The primary research aim for this paper is to identify how guide dog puppy-raisers engage in and perceive preparedness for their guide dog puppies. We focus on college students as a population of puppy raisers because college students are at a unique developmental phase in which perceptions of risk and safety, as well as independence, begin to take on a new role in their lives [55]. Additionally, it is important to understand the ways in which college students prepare for hazards [56]. If being a guide dog puppy volunteer is associated with lower or higher levels of preparedness, this has important implications for higher education institutions in terms of planning and outreach. No studies exist that examine the ways in which college students as puppy-raisers prepare for disasters. Studies so far on dogs on college campuses focus on dogs as stress relief for students [57,58] rather than how having a guide dog puppy in training changes the students’ level of preparedness.

## 2. Method & Materials

### 2.1. Participants

The researchers gathered data using a Qualtrics survey platform to reach volunteer puppy raisers within a southeastern college town. Qualtrics is an electronic survey platform that collects responses and can be automatically uploaded through the internet or in which responses can be uploaded onto a hard drive. The participants were contacted through an email list serve composed of the guide dog puppy raisers. There were approximately 300 people on the list-serve. The puppy raisers targeted for this survey are volunteers through an organization based in the northeastern United States that recruits’ college students throughout the US for puppy-training. Specifically, as part of their puppy-raising program, this school has volunteer groups throughout the east coast, with a lot of groups in southeastern states. One of the researchers for this project knew the leader of this foundation from her personal social network, which enabled the research team to distribute the survey through the email list serve. In other words, the recruitment was through convenience sampling [59] but with the support of the leader of this organization that trains guide dogs. While richness of data is beneficial from open-ended interviews [60], we chose a survey format because it facilitates rapid data collection and is convenient for respondents to take a web-based survey rather than coming to a specific location at a specific time for data collection.

Because of the nature of being a puppy raiser (i.e., taking care of a young puppy, taking it with you everywhere you go), groups usually center around universities, as many of their volunteers are college students. Although the guide dog school is headquartered in the northeast, they have a southeastern coordinator who maintains the southern groups. A reminder email was sent two weeks after the initial survey recruitment and notification.

### 2.2. Procedure

The survey (Appendix A) was conducted under a university Institutional Review Board in December 2017 and January 2018. The survey link contained a consent form, an introduction to the study and allowing participants to consent and continue on with the questionnaire. The responses were automatically and anonymously uploaded into the Qualtrics system after each submission and transferred to SAS statistical software for data cleaning and analysis.

### 2.3. Data Analysis

The results from the Qualtrics survey had 84 responses, not including pilot responses. Analyses were conducted on the sample size of *N* = 84 (response rate = 28%), but there were missing data for 31 respondents. Of those 31, 84% did not answer past the first two questions and were not included in the final analyses, and the remaining 16% left partial responses and were included. There were a total of 53 respondents who completed the entire survey. SAS version 9.4 was used to code the survey responses, assess frequencies, and conduct chi-square analyses.

## 3. Results

Out of the 53 participants who completed the entire survey, the majority were female (*n* = 48, 91%) with a mean age of 28 years (*SD* = 14). The respondents were predominantly White (*n* = 47, 88%). The remaining respondents were Hispanic (*n* = 3, 1%), Asian (*n* = 2, <1%), and African American (*n* = 1, <1%). 40 out of the 58 participants identified as college students. Seniors made up most of the college students (*n* = 19, 48%), followed by graduate and dual-degree students (*n* = 9, 22%), juniors (*n* = 8, 20%), and sophomores (*n* = 4, 10%). It is likely that the high number of responses from college students is because most of the people on the list-serve are students and because they may be more likely to check email frequently because of schoolwork and communication. There were no freshman participants; however, it is less likely for first year students to be puppy raiser volunteers as it takes time to learn about the program, apply, and go through the volunteer process. Among the puppy raisers, 65% of them were repeat raisers, meaning they had already raised one or more dogs, while 35% were first time raisers.

The first items in the survey measured evacuation plans, locations, and logistics (Refer to Appendix A). Among the puppy raisers who said they had an evacuation plan for their puppy in training (*n* = 17), many stated they would just put the dog in the car with them and leave. Some raisers specifically added they would bring food, water, toys, a dog bed, or a crate (*n* = 8). Only two raisers said anything about bringing a preparedness or disaster kit for their dog, and another two mentioned bringing papers and vaccine records. One survey item inquired about where participants would go if needing to evacuate. Many respondents (*n* = 33) stated they would go stay with family, including parents, siblings, or even extended family. Family was also a popular response among college students when asked where they would go if they had to evacuate their parents’ house.

Four survey questions specifically asked participants about plans relating to Tropical Storm Irma of 2017 since the storm had hit southeastern states a few months prior to the survey. An analysis of the crude relationship between puppy raiser experience (first time raiser versus repeat raiser) and consideration to evacuate for Irma gave an odds ratio of 15.3. This reflects that the odds of first-time puppy raisers considering evacuation for Irma was 15.3 times the odds of repeat puppy raisers, revealing that first-time raisers were much more likely to consider evacuation, *X*^2^ (1, *N* = 39) = 11.37, *p* < 0.05. Also, of the 33 participants (77%) that said they prepared supplies specifically for their animal before Irma, 32 (74%) mentioned essentials such as food and water, 4 (9%) said they gathered medical supplies or a first aid kit for their animal, 5 (12%) mentioned outside gear for the dogs when going to the bathroom or a way for the dogs to go to the bathroom inside, and a final 2 (5%) mentioned getting together papers or vaccine records. It should be noted that some respondents were included in more than one category as they mentioned more than one of the above answers.

In response to the item inquiring about having a disaster kit at home, over half (58%) of the participants said no. Most participants (82%) said that their companion animal or puppy in training makes them feel safe daily as opposed to the 2% who said their companion animal or puppy in training makes them feel unsafe. Table 1 shows frequencies of the risk perception responses, reflecting what participants view as the highest risk to their companion animal or puppy in training daily versus what they view as the highest risk to themselves.

## 4. Discussion

The primary aim of this study was to explore levels of preparedness and risk perception in college students with service dogs (puppies) in training. We expected volunteer raisers who had raised more than one dog to be more prepared for sudden emergencies or disasters (i.e., have a disaster kit, have a plan for their dog). However, over half of the puppy raisers surveyed did not have a disaster kit of any kind and although many prepared food and water for their dog prior to Irma, very few mentioned vaccine records or anything medical-related for the dog. Additionally, first-time puppy-raisers were more prepared than people who had raised multiple puppies. This may be because of the influence of having a “new animal” and potential feelings of protectiveness, like findings in which families with children are more likely to have higher levels of household preparedness [61].

Vaccine records are extremely important to have on hand in case of an emergency as they would be needed when taking an animal to a boarding facility, vet office, or even a shelter [62]. Being turned away at any of those facilities due to lack of records can also play a role in people not wanting to evacuate, especially when they are unable to locate a pet-friendly shelter to begin with [48].

In response to the item inquiring about having an evacuation plan for their puppy in training, over half stated not having a plan at all. Among those that did have a plan, the majority just mentioned evacuating with the dog in the car. Two participants mentioned bringing a crate, and although research suggests transportation challenges as the main cause of evacuation failure for companion animal guardians, it should be noted that these dogs are trained to ride in the footwell of cars (not in crates) and that every puppy raiser should be issued a crate when they receive their puppy in training. Again, few respondents mentioned vaccine records or a disaster kit, with only one participant stating they always kept a kit in the car for their dog. These results support existing literature on preparedness among college students in general, with the most common preparedness activities found to be storing water or food, having a working flashlight, and taking a Cardiopulmonary Resuscitation (CPR) class [63]. In a similar way that students take few protective actions because they assume the university will take care of them in the event of a disaster, puppy raisers may expect the same resources and guidance from the organization for which they volunteer [42].

Many participants said in the event of an evacuation, they would evacuate to a family or relative’s place of residence. No specific logistical issues were mentioned; however, that may be due to the socioeconomic status of the survey sample. Most respondents have access to cars—the dogs have been trained and accustomed to riding in a car—and therefore most participants would not need the use of local shelters. Previous hurricane evacuation data reflects that most evacuees stay with relatives or friends [64]. Many college students mentioned evacuating to where their parents live and is important to note since students may in some cases rely on parents for risk-related information more so than media or other interpersonal sources [44].

There was a difference in likelihood of evacuating or considering evacuating prior to Irma between first time raisers and repeat raisers, with first time raisers more likely to do so. This could be due to repeat raisers being more confident about their responsibility towards their puppy in training as they have been through the raising process before. Of the college student participants, many reported being enrolled in majors relating to animals or other involved sciences. The confidence among puppy raisers concerning the safety of their dog could also stem from their area of study being closely related to the risk, a reason many college students tend to have low risk perception [32,33]. It should also be noted, again, that the samples for this study were not diverse in terms of sex and ethnicity, which creates limitations for the implications regarding additional analyses to compare risk perception and preparedness among respondents. Additional research in the area of guide dog puppy raisers should include a sample that has more ethnic and gender diversity, especially since minority groups more susceptible to adverse outcomes in disasters [65].

Many of the participants stated that their companion animal or puppy in training made them feel safe. These results are consistent with other findings that suggest pets help enable a sense of security and serve as a haven and secure base for their guardians [66,67]. This sense of safety from the human-animal bond also suggests that companion animal guardianship can serve as a protective factor during evacuations where guardians are more likely to get to safety for the sake of their companion animal [68]. College students also directly benefit from pets, where just a simple twenty-minute canine therapy session reduced stress and even homesickness [69].

However, despite the sense of security the puppies in training and pets bring to their guardians, there are still daily risk perceptions that differ from person to person. Table 1 reflects that most participants viewed the highest risk to themselves and their companion animal to be an auto accident. Despite the number of young drivers involved in car accidents every year, some participants viewed the highest risk to their companion animal to be an event of lower probability, such as severe weather, structural fire, or an active shooter [36]. Because individuals have less control over these types of events, it is possible that fear caused a heightened risk perception despite an auto accident being a more significant risk [70].

## 5. Conclusions

This study took place in the United States and therefore the results are not generalizable beyond the specific population of college students with guide dog puppies. Also, as we mention in the broader discussion, the homogenous sample (white, able-bodied, and female) is a limitation. Future research should include diverse groups of volunteers. Additionally, the small sample size of survey responses is a limitation of this study. Although there were differences between first time and repeat puppy raisers, the small sample did not yield more than one statistically significant finding. Future research should also focus on the potential difference in preparedness practices and activities between companion animal guardians and puppy raisers since puppy raisers take their dog most places with them as opposed to a companion animal guardian. Future research should also include in-depth qualitative data in which students as puppy-raisers share their thoughts on safety, risk, and preparedness. This would add richness to the data regarding motivations, perceptions, and processes for preparedness. This might also uncover new information about levels of bonding with their guide dog puppies and how levels of bonding relate to preparedness.

Finally, it is also important for future research to explore the issue of branding or the “trendiness” of having a guide dog puppy among college students, especially in the context of understanding how this experience is different for the person training the dog compared to the person who is ultimately paired with and requires the dog because of a functional-access need. The college students in this sample were white and able-bodied [71] whereas this may not be the case for functional access individuals in which having a service dog may be another form of labeling the person as “disabled” [72,73] and in which preparedness may entail different meanings and processes. In other words, it may be important to understand how the accompaniment of a service-dog elicit stigma from others and the ways in which this impacts functional and access needs individuals in hazards events.

The results from this study can be used as a foundation in future research for service dog organizations into hazard and disaster preparedness among their puppy raiser volunteers. Because service dog volunteer groups are sometimes spread several states away from the headquarters of the guide dog organization, accountability and tracking the dogs could become difficult in the event of an emergency or disaster. For some volunteer groups, there are hierarchical leadership roles in place to help keep track of the raisers and all the dogs. These leadership positions could incorporate potential preparedness protocols into their volunteer group meetings or even just going over what to include in a pet-related disaster kit. Since most groups are required to attend monthly meetings, those are already designated times when many puppy raisers and volunteers come together where preparedness practices could easily be discussed among the group.

Additionally, since having a service dog in training made students feel safer, this information could be used in developing recruitment messages for new volunteers. Having service puppies in training makes students feel safer and possibly less stressed. A key message for college student volunteers could be something along the lines of, “Your service puppy in training helps you to feel safe. Learn how to keep your puppy in training safe during disasters”. In general, puppy raisers are a distinct and dynamic group of companion animal guardians that are most likely not accounted for in many companion animal management plans, so it is important to practice individual and community-level preparedness for the safety of both people and animals. Finally, it is also important to consider that preparedness strategies and meanings will likely be different for trainers than they are for people who are later paired with the dogs and this may impact implications for experiences of both the dogs and humans. This discrepancy should be accounted for when organizations are creating training protocols for raising guide dog puppies (i.e., considering a more inclusive and aggressive outreach of puppy-raisers who have functional-access needs, people of color, and more diverse groups of people). This is also reflected as a guideline for best practices in preparedness and disaster planning [74].

## Figures and Tables

**Table 1 animals-10-00246-t001:** Survey responses to “highest risk to self” and “highest risk to pet”.

Risk to Pet			Risk to Self		
*Risk*	*Frequency*	*Percentage (%)*	*Risk*	*Frequency*	*Percentage (%)*
Auto Accident	32	74.4	Auto Accident	43	74.1
Canine Flu/Illness	5	11.6	Flu/Illness	10	17.3
Severe Weather	2	4.70	Severe Weather	1	1.70
Active Shooter	1	2.30	Active Shooter	1	1.70
Structural Fire	2	4.70	Structural Fire	0	0
Other	1	2.30	Sport/Physical Injury	2	3.50
			Other	1	1.70

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
