# Peer review of "Disaster Preparedness among Service Dog Puppy- Raisers (Human Subject Sample)"

_animals, 2020, doi:10.3390/ani10020246_

Round 1
Reviewer 1 Report
This is a long winded study which says very little that is not common knowledge. Although safety of puppy walkers dogs due to a physical accident should indeed be included with, I would hope, there much more extensive training on how to handle and teach their puppies who spend the best part of their young life with these people and are extremely sensitive at this age to how handled and taught which will effect the whole of their lives thereafter, I would not have though it should need a long, dull paper to point out! The dangers to the dog are far greater from being wrongly handled, taught or used simply as a "safety" measure or to make new friends among other humans, than that the dog will be injured in a physical accident.
In other words unless this paper is reduced to a short paragraph pointing with a note to ensure that the trainers of the dog walkers include some advise, I see no reason to publish it.
Rewriting the paper with an emphasis on all the mental hazards the puppy is exposed to in his short life changing places, carers, trying to make and keep good emotional relationships etc is the real risk to the puppies and why so many are finally rejected from becoming guide dogs. This would make a good paper.
"therapy for humans" what about the dog line 138-0. Meaning line 150-51 rewrite
172 risk perception only physical risk
184. These students just use texts more than they used to, not "being taken care of by university"!!
A medical kit will be much the same for a dog or a human.
320 now deny what previous said in 184. What about risks to puppy such as getting lost or separated those with whom bonded, being run over, bored etc etc???
Author Response
We thank the reviewer for their comments and feedback. We address your points below.
First, the reviewer marked that methods could be improved but did not provide a specific example of ways in which to improve the methodology. Therefore, we have maintained the current format for the methods section.
Additionally, the reviewer's comments reflect a desire for the entire foundation of the paper to be re-written. To be clear, the disaster preparedness of puppy-raisers increases safety of dogs, which is why we carried out this study. As we state on lines 62-68: "Less attention has been given to the topic of service animals in disasters and the ways in which people who rear service animals engage in disaster preparedness. In the United States, college students are increasingly volunteering to train guide-dog puppies—and many of the campuses in which they live are prone to hurricanes and other hazards. This research is an attempt at understanding the ways in which guide-dog trainers prepare their puppies for disasters and emergency scenarios. Specifically, this research focuses on the emerging population of college student puppy raisers and the ways in which they perceive preparedness, safety, and risk for themselves and their puppies in-training."
The point of this paper is not to argue the acceptability of the practice of dogs reared for service work, but rather to understand how the dogs can be protected from harm in the event of hazards through increased preparedness of the person/volunteer who is training the puppy.
We slightly re-worded phrasing for clarity on lines 151 and 172.
The reviewer's point about line 184 is in reference to the Lovecamp et al study, not our own research. We slightly re-worded the sentence on line 320 to avoid the appearance of contradicting what we referenced on line 184 (again, these are in reference to other studies - Koskan et al, 2012).
Finally, for your last comment- we acknowledge that guide-dogs in training may indeed become bored at times, just like other dogs who are long-term companion animals. Again, our paper is not designed to argue the merits of the practice of training service dogs. Understanding disaster preparedness has the opportunity to increase safety of the dogs in emergencies.
Reviewer 2 Report
I think the work is quite ready for publication. I have only one small suggestion to improve the bibliography (see below).
Line 91:”Despite working dogs serving a different role than companion animals, they may have similar attachment levels to their handlers as pets do to their guardians [13]. Assistance 92 dogs serve important roles for their handlers, especially for handlers who cannot see or are visually 93 impaired [12].”
Ther is now scientific evidence about the correlation between the Owner’s
Attachment Profile and the Owner-Dog Bond. The following paper should be cited at this point:” Siniscalchi M, Stipo C, Quaranta A (2013) "Like Owner, Like Dog": Correlation between the Owner’s Attachment Profile and the Owner-Dog Bond. PloS ONE 8(10): e78455. doi:10.1371/journal.pone.0078455
Author Response
We thank reviewer 2 for their comments. We have added the Siniscalchi and Quaranta (2013) citation to line 91.
Reviewer 3 Report
Today, it is well recognized throughout the world that companion animals of a variety of species provide direct health benefits, social well-being, comfort, and stress, crisis and disaster management both for the humans and the animals. While the Pets Evacuation and Transportation Standards (PETS) Act of 2006 delineated an important policy for creating the plans for managing pets in disasters, preparedness for individuals during disasters remains extremely important for all companion animal guardians, service dog users, and those with service dogs in training.
The current web-based survey was aimed at addressing this point, namely, to understand disaster preparedness among service dog puppy raisers. There were 53 completed survey responses (a 28% response). 59% of those who had an evacuation plan for their puppy in training stated they would put the dog in their vehicles for evacuating to safety in the event of a hurricane or other disaster. Interestingly, the odds of first-time puppy raisers who considered evacuation for Hurricane Irma in 2017 was 15.3 times that of repeat raisers. Despite this fact, more than half the raisers reported that they did not have a disaster kit. Most respondents (82%) indicated that having a service puppy in training makes them feel safer.
The findings of the study, despite the small sample size, are well-described and documented. A strength of this manuscript is the detailed discussion of the study limitations. Of particular concern is that the respondents were all white, able-bodied females. Clearly, as the authors indicate, a more diverse population group from various ethnicities, both sexes, other countries and with disabilities also needs to be studied, to solidify or refute the current conclusions.
Another important, yet often overlooked, point raised by the authors was the need to have vaccine records on hand in case of an emergency, when taking an animal to a boarding facility, vet office, or even a shelter. The same could be said for having a summary of health and any current or recent veterinary treatment records along with the use of heartworm, flea and tick preventives. The authors should add these additional points to their Discussion.
Language Comments. Line 96. Do you mean "origins"?
Line 370. Do you mean "elicit" ?
Author Response
We thank the reviewer for their comments and feedback.
Please note that we changed the wording on lines 96 and 370 as you suggested.